# Are *ENT1*/ENT1, *NOTCH3,* and miR-21 Reliable Prognostic Biomarkers in Patients with Resected Pancreatic Adenocarcinoma Treated with Adjuvant Gemcitabine Monotherapy?

**DOI:** 10.3390/cancers11111621

**Published:** 2019-10-23

**Authors:** Lucie Jiraskova, Ales Ryska, Erik Jurjen Duintjer Tebbens, Helena Hornychova, Filip Cecka, Frantisek Staud, Lukas Cerveny

**Affiliations:** 1Department of Pharmacology and Toxicology, Faculty of Pharmacy in Hradec Kralove, Charles University, 500 05 Hradec Kralove, Czech Republic; jiraskl2@faf.cuni.cz (L.J.); frantisek.staud@faf.cuni.cz (F.S.); 2The Fingerland Department of Pathology, Faculty of Medicine and University Hospital Hradec Kralove, Charles University, 500 05 Hradec Kralove, Czech Republic; ales.ryska@fnhk.cz (A.R.); helena.hornychova@fnhk.cz (H.H.); 3Department of Biophysics and Physical Chemistry, Faculty of Pharmacy in Hradec Kralove, Charles University, 500 05 Hradec Kralove, Czech Republic; jurjend@faf.cuni.cz; 4Department of Surgery, Faculty of Medicine and University Hospital Hradec Kralove, Charles University, 500 05 Hradec Kralove, Czech Republic; filip.cecka@fnhk.cz

**Keywords:** adjuvant gemcitabine monotherapy, equilibrative nucleoside transporter 1, neurogenic locus homolog protein 3, miR-21, resected pancreatic ductal adenocarcinoma, prognostic biomarker

## Abstract

Evidence on equilibrative nucleoside transporter 1 (ENT1) and microRNA-21 (miR‑21) is not yet sufficiently convincing to consider them as prognostic biomarkers for patients with pancreatic ductal adenocarcinoma (PDAC). Here, we investigated the prognostic value of *ENT1*/ENT1, miR-21, and neurogenic locus homolog protein 3 gene (*NOTCH3*) in a well-defined cohort of resected patients treated with adjuvant gemcitabine chemotherapy (*n* = 69). Using a combination of gene expression quantification in microdissected tissue, immunohistochemistry, and univariate/multivariate statistical analyses we did not confirm association of *ENT1*/ENT1 and *NOTCH3* with improved disease-specific survival (DSS). Low miR-21 was associated with longer DSS in patients with negative regional lymph nodes or primary tumor at stage 1 and 2. In addition, downregulation of *ENT1* was observed in PDAC of patients with high *ENT1* expression in normal pancreas, whereas *NOTCH3* was upregulated in PDAC of patients with low *NOTCH3* levels in normal pancreas. Tumor miR‑21 was upregulated irrespective of its expression in normal pancreas. Our data confirmed that patient stratification based on expression of *ENT1*/ENT1 or miR‑21 is not ready to be implemented into clinical decision-making processes. We also conclude that occurrence of *ENT1* and *NOTCH3* deregulation in PDAC is dependent on their expression in normal pancreas.

## 1. Introduction

Pancreatic ductal adenocarcinoma (PDAC) is one of the most aggressive solid malignancies, representing the fourth leading cause of cancer-related mortality in the modern world [1]. Nearly as many people die of the disease as are diagnosed each year [2], and by 2030, PDAC is expected to be the second leading cause of cancer-related death [3]. The main treatment for potentially curative therapy is still surgical removal of the tumor with tumor-free resection margins (R0). Resection is achievable in less than 20% of patients [4] and is associated with 10% and 7.7% five- and ten-year survival, respectively [5,6,7].

Adjuvant gemcitabine monotherapy (GEM) doubles the five-year overall survival to up to 21% in patients with R0 resection [5,6,7,8]. Recently, Neoptolemos et al. (2017) published results of a multicenter, open-label, randomized phase clinical trial (ESPAC-4) which demonstrated superiority of GEM plus capecitabine combination over monotherapy with GEM: the overall survival was 28 months in the GEM with capecitabine group vs. 25.5 months in the GEM monotherapy group. However, a higher frequency of grade 3–4 adverse events has been reported for GEM plus capecitabine combination [9]. Another multicenter, open-label, randomized phase III clinical trial (PRODIGE 24/CCTG PA.6 (NCT01526135) showed that modified-dose FOLFIRINOX (mFOLFIRINOX), consisting of oxaliplatin at 85 mg/m^2^, leucovorin at 400 mg/m^2^, irinotecan at 150 mg/m^2^, and 5-fluoroucil at 2.4 g/m^2^ increases three-year survival compared with GEM monotherapy (63.4% vs. 48.6%). Nevertheless, the safety profile of the mFOLFIRINOX regimen was suggested as less favorable than that based on gemcitabine monotherapy [10].

Besides showing more acceptable toxicity, it has been documented that GEM monotherapy is a cost-effective option when compared with GEM plus capecitabine in an adjuvant regimen [11]. However, to improve the cost-effectiveness of adjuvant GEM monotherapy, it is important to identify patients that could significantly profit from the treatment [12,13,14,15].

Over the last decade, there has been a hunt for valuable prognostic/predictive biomarkers and reliable methods for their analysis that could be helpful in the estimation of PDAC patients’ responsiveness to GEM. Of the biomarkers investigated so far, human equilibrative nucleoside transporter 1 (ENT1), microRNA-21 (miR-21), and neurogenic locus homolog protein 3 (NOTCH3) are considered promising.

ENT1 is the most important transporter for GEM influx into pancreatic cells [16,17], and hence has been extensively investigated. Currently, some evidence from immunohistochemistry and mRNA analyses performed in formalin-fixed paraffin-embedded (FFPE) samples supports a hypothesis that low expression of ENT1 might be an indicator of chemoresistance to GEM in resected patients [12,13,18,19], but contradictory findings have also been published [20,21].

miR-21 is a short (22 nt), very stable, noncoding RNA targeting *Bcl-2* [22] that likely plays an important role in preventing apoptosis, thus functioning as a proto-oncogene [23]. High miR-21 expression has been associated with significantly shorter overall survival in resected patients [24,25].

To date, most studies investigating the prognostic value of ENT1 and miR-21 have been conducted in a small cohort [19], on patients treated with a combination of adjuvant GEM and radiation [12,19,26], a cohort mixing patients with adjuvant and palliative settings [25] and/or for which chemotherapy is not reported [27]. Only a limited number of studies have been performed in a well-defined homogenous cohort of resected patients with adjuvant GEM monotherapy [13,28]. The overall evidence on ENT1/*ENT1* and miR-21 is thus encouraging, but not yet sufficiently convincing to implement this procedure in the clinical environment.

NOTCH3 is linked to the GEM-resistant PDAC phenotype. NOTCH3 confers cell extracellular interactions, such as invasion, migration, motility, and modification of survival of pancreatic cells [29]. NOTCH3 is related to GEM-induced caspase-mediated apoptosis [30]. Using multivariate analysis, high *NOTCH3* mRNA levels have been associated with shorter survival of GEM-treated patients with advanced PDAC [31]. However, to date, this biomarker has not been evaluated in resected patients with GEM monotherapy.

Several studies have indicated that low miR-21, low NOTCH3, and high *ENT1* may be used as GEM-independent favorable prognostic factors of the effect of GEM therapy [26,32,33,34]. Considerable inter-individual expression of NOTCH3 and miR-21 in tumor tissue, ranging from negative to strongly positive, has been reported [32,35], and increased expression of these molecules in PDAC has been suggested [25,27,32,33,35,36,37]. However, upregulation does not appear in all patients and it remains to be elucidated whether elevated levels of these molecules correspond with either low or high expression in normal pancreas or are independent. Moreover, data about expression of *ENT1* in PDAC compared with normal pancreas are completely lacking.

In this study, we aimed to use quantitative real-time polymerase chain reaction (qRT-PCR) analysis of *ENT1*, miR-21, and *NOTCH3* expression in FFPE samples collected from a homogenous group of patients with resected PDAC, treated with adjuvant GEM therapy (*n* = 69) to evaluate the prognostic value of the associated transcripts for the estimation of disease-specific survival (DSS). Moreover, we analyzed expression profiles of *ENT1*, miR-21, and *NOTCH3* in PDAC tissue of different patient subgroups, defined by the median of expression in normal pancreas.

## 2. Results

### 2.1. Clinical–Pathological Characteristics of Patients

Clinical characteristics, including age, gender, surgery type, resection margin status, stage of primary tumor, regional lymph nodes, distant metastasis, DSS, and American Society of Anesthesiologists (ASA) score, of the patients (*n* = 69) are summarized in Table 1. Thirty five (50.7%) patients finished all the cycles of chemotherapy, whereas 34 patients (49.3%) prematurely terminated treatment because of disease progression (14; 20.2%), toxicity (17; 24.6%), heart failure (1; 1.5%), respiratory failure (1; 1.5%), or sudden death (1; 1.5%). In the monitored cohort, one patient had small metastases in the peritoneum in close proximity to the pancreas; the metastases were surgically removed. Fourteen patients were alive at the end of follow-up (31st December 2018).

### 2.2. Clinical–Pathological Factors and Chemotherapy Response as Survival Markers

To evaluate the effects of tumor and patients’ characteristics and type of resection on DSS, we performed a statistical analysis using Kaplan–Meier curves and the log rank test. We dichotomized the tested cohort based on resection margin status (R0/R1), presence of metastatic involvement of regional lymph nodes (N0/N1), primary tumor stage T(1,2)–T(3,4), ASA score ASA(1,2)/ASA(3,4), patients’ age (>65 age/<65 age), gender, and type of resection (PD/DP). R0 was found to be associated with significantly longer DSS over R1 (21 months vs. 14 months, *p* = 0.0314, hazard ratio = 0.5663, and 95% confidence interval = 0.3062–1.047) (Figure 1A), whereas other analyzed parameters did not show any association with DSS (Figure 1).

### 2.3. Analysis of ENT1, NOTCH3 mRNA, and miR-21 Levels in PDAC Tissue and Their Association with DSS

To investigate possible association of *ENT1*, *NOTCH3*, and miR-21 levels with patients’ DSS, we quantified gene expression of the transcripts in tumor tissue microdissected from FFPE samples (*n* = 69). For the purpose of Kaplan–Meier survival analysis, patients were dichotomized by the median of expression of the respective molecules into two groups (low <50 and high >50%). However, we did not observe any association between *ENT1*, *NOTCH3*, or miR-21 expression and patients’ DSS (Figure 2). Only patients with low miR-21 showed a trend for longer DSS (22 vs. 16 months, *p* value 0.4649, Figure 2C). Subsequently, patients were divided based on the expression of individual markers into quartile subgroups. We selected only subgroups of patients with the lowest (<25%, first quartile) and highest (>75%, fourth quartile) expression of each marker for subsequent analyses. However, despite using these more polarized subsets of patients, no significant differences were observed (Figure 2D–F).

In multivariate DSS analysis adjusted to resection margin status (R0/R1), gender (female/male), ASA score (I–III), primary tumor stage (T1–T4), regional lymph node (N0/N1), and type of resection (PD/DP), we did not observe any significant association between patients’ DSS and mRNA expression of *ENT1* (Table 2), *NOTCH3* (Table 3), and miR-21 (Table 4). As shown in Table 4, only high levels of miR-21 tended to statistical significance (*p* = 0.089, hazard ratio = 0.475), and T3 was shown as a favorable factor (*p* = 0.036, hazard ratio = 0.085).

### 2.4. Immunostaining of ENT1 in FFPE Samples of PDAC

Immunohistochemical evaluation is the most frequently used procedure for evaluating ENT1 as a prognostic/predictive biomarker for patients with PDAC. Therefore, it is considered as a “standard” method. ENT1 staining showed predominantly membranous positivity in the cells of Langerhans islets and lymphocytes. Thus, the presence of this type of staining in normal Langerhans islets served as an internal positive control of the method (Figure 3). Quantitative scoring using light microscopy was conducted by a single experienced pathologist (AR). Of the 63 tissue samples, 54 had detectable ENT1 immunostaining (intensity score from 1 to 3), some of which revealed heterogeneous expression with regions lacking ENT1. Nine samples of PDAC were without any detectable ENT1 expression (intensity score 0). The percentage of adenocarcinoma cell staining at each intensity level was recorded for each specimen. Patients with a high histoscore (values 6–9) for ENT1 (*n* = 24) revealed a median DSS of 23 months, whereas patients with a low (values 1–5) histoscore (*n* = 30) and negative (*n* = 9) ENT1 showed a median DSS of 18 months (Figure 4).

In the multivariate model adjusted to resection margin status (R0/R1), gender (female/male), ASA score (I–III), primary tumor stage (T1–T4), regional lymph node (N1/N0, i.e., positivity/negativity), type of resection (PD/DP), and ENT1 protein expression analyzed by immunohistochemistry, we observed significant positive association between patients’ DSS and N0 (*p* = 0.049, hazard ratio = 0.424; Table 5). The R0 resection margin status also revealed a trend for positive association with patients’ DSS (*p* = 0.051, hazard ratio = 0.505; Table 5).

### 2.5. Analysis of ENT1 Transcripts in Subgroups with Negative, Low, and High ENT1 Protein Expression Analyzed by Immunohistochemistry

Using absolute qRT-PCR analysis, we compared the number of transcripts in samples of PDAC, stratified as tumors with no, low, or high protein expression. Despite an increasing trend from negative to high subpopulations, we did not observe any significant differences in mRNA expression among these groups of samples (Figure 5).

### 2.6. Analysis of Patients’ DSS Association with ENT1, NOTCH3 mRNA, and miR-21 Levels in Patients’ Subgroups, Defined by Clinical–Pathological Characteristics

As we hypothesized that our data might be affected by a high proportion of patients with positive resection margin status (R1), shown to be associated with a significantly shorter DSS (Figure 1A), N1, and/or more advanced primary tumor (T3,4), we analyzed the effect of expression of selected molecules separately in subgroups categorized as R0, N0, or T(1,2) and subsequently, subgroups R1, N1, or T(3,4). In the N0 subgroup, we observed a significantly improved DSS in patients expressing low levels of miR‑21 (<50%) over those expressing high levels (>50%) of miR‑21 (48 months vs. 15 months; *p* = 0.0308; hazard ratio = 0.3706; 95% confidence interval = 0.1139 to 1.205) (Figure 6A). Comparable to the T(1,2) subgroup, patients with low miR‑21 expression (<50%) demonstrated longer DSS than those with high miR-21 expression (>50%) (29 months vs. 10 months; *p* = 0.0438; hazard ratio = 0.3341; 95% confidence interval = 0.09289 to 1.255) (Figure 6B). In other cases, we did not find any correlation between the expression of any tested molecule and patients’ DSS.

### 2.7. Quantitative RT-PCR Analysis of ENT1, NOTCH3 mRNA, and miR-21 Expression in FFPE Samples of PDAC

Gene expression of selected markers *ENT1*, *NOTCH3*, and miR-21 was determined in both tumor and normal tissue. This experiment was performed in samples collected from 65 patients, for whom we had FFPE blocks containing sufficient amounts of both tumor and normal tissue. Considerable expression variability was observed for all the analyzed transcripts in both tumor and normal pancreas: *ENT1* (2.4 and 2.6 logs for tumor and normal pancreas, respectively), *NOTCH3* (2.8 logs for both types of tissues) and miR-21 (1.7 and 2.1 logs for tumor and normal pancreas, respectively). Using nonparametric unpaired Mann–Whitney test we found significantly reduced overall *ENT1* mRNA expression in tumor tissue compared with normal pancreas (Figure 7A): decreased *ENT1* expression was detected in tumor tissue of 67.7% (44/65) patients. When analyzing medians of *NOTCH3* and miR-21 expression, both were significantly increased in tumor tissue: upregulation was observed in 72.3% (47/65) and 95.4% (62/65) of patients, respectively (Figure 7B,C). Because we hypothesized that the extent of *ENT1*, *NOTCH3*, and miR-21 deregulation in PDAC might be dependent on their levels in corresponding normal pancreas, we divided patients based on the median of expression of each marker in healthy pancreas (high >50%; low <50%) and compared the expression of each marker in tumor and normal pancreas in these subgroups in paired fashion. Wilcoxon matched-pair signed-rank test showed significantly decreased *ENT1* in PDAC of the high (>50%) subgroup. *NOTCH3* showed significant upregulation in tumor tissue of the low (<50%) subgroup specimens only, and the amount of miR-21 was increased in both tested subgroups. Subsequently, Mann–Whitney test demonstrated that overall *ENT1* expression in PDAC was still significantly higher than that found in normal pancreas of the low (<50%) subgroup (Figure 7D). *NOTCH3* expression in tumor tissue of the low (<50%) subgroup was lower compared with normal pancreas in the high (>50%) subgroup (Figure 7E). Levels of miR-21 in PDAC of the low (<50%) subgroup were comparable with those in normal pancreas in the high (>50%) subgroup (Figure 7F).

## 3. Discussion

When considering the economic aspects of healthcare interventions [38], adjuvant GEM monotherapy of PDAC represents a cost-effective option [11]. However, the cost-effectiveness of this regimen can potentially be further improved by identification of biomarkers for personalized GEM administration [39]. Considerable attention has been devoted to ENT1 [12,13,14,15,18], whereas, to the best of our knowledge, there have only been two studies on miR‑21 [24,25] and none on *NOTCH3*.

Immunohistochemistry analysis is an essential tool in everyday clinical practice. However, the performance of this method depends on the availability of a high-quality antibody [40,41] and experienced pathologist [42,43,44,45]. Further, the heterogeneity of outcomes obtained with different types of anti-ENT1 antibody [15,21,46] and lack of established standardized scoring procedure for evaluation of ENT1 expression [47] represent critical obstacles preventing full adoption of ENT1 analysis into clinical practice [20]. When staining with antibody 10D7G2, a correlation between high ENT1 expression and improved survival of PDAC patients with adjuvant GEM treatment was shown [18,19,46,48], whereas staining with SP120 antibody failed to reveal any such correlation [20,21]. Recently, Kalloger et al. (2017) attempted to explain the differences between these antibodies. Using a unique statistical approach, they concluded that both antibodies are suitable for stratification of patients but, surprisingly, SP120 is the more useful [28], making the issue of ENT1 evaluation in PDAC even more complicated. In contrast, qRT-PCR analysis might offer a more straightforward method for decision-making. Moreover, qRT-PCR allows quantification of multiple molecules in parallel. However, the applicability qRT-PCR analysis outcomes may be hampered as levels of transcripts may not be proportionally reflected by protein amounts and/or tumor and normal tissue differ only in the subcellular distribution of protein but not the total amount. Moreover, consensus about how to stratify patients based on *ENT1* gene expression and miR-21 levels has not been established: groupings based on (i) tertile of expression [19], median of expression [25], or (iii) results of recursive descent partition analysis have been suggested [12,26]. Despite these methodological drawbacks, high levels of *ENT1* mRNA detected by qRT-PCR in FFPE samples have been demonstrated to be a favorable prognostic marker in PDAC patients with adjuvant GEM therapy [12,19].

Contrary to published data [12,19,24,25], we did not demonstrate in our cohort, strictly defined by administration of the recommended adjuvant GEM regimen, that high *ENT1* and low miR-21 were favorable prognostic factors. We also did not observe an association of low *NOTCH3* with improved DSS (Figure 2). However, this latter finding corresponds with results obtained using univariate analysis performed in patients with advanced disease [31]. Subsequent multivariate analyses confirmed our observation that *ENT1*, *NOTCH3*, and miR-21 are not prognostic biomarkers of patients’ responsiveness to GEM (Table 2, Table 3 and Table 4). However, when we divided the patients into two subgroups based on clinical–pathological characteristics, univariate analysis revealed that low miR‑21 is a favorable prognostic factor in N0 patients, as previously demonstrated in another cohort with unspecified post-surgery treatment [32] and also in T(1,2) patients (Figure 6).

As our data on *ENT1* mRNA are in conflict with previous reports [12,19], we also performed an immunohistochemical analysis. DSS in patients with a high histoscore was 23 months, compared with 18 months in patients with low/negative staining, which is comparable to reported data [13]. However, in accordance with the analysis of *ENT1* mRNA, high levels of ENT1 protein were not shown to be a favorable prognostic factor (Figure 3, Table 5). This is in contrast with previous studies obtained using 10D7G2 antibody [13,14,18,28]. Only N0 was shown as an independent factor (*p* = 0.049), and R0 almost reached statistical significance (*p* = 0.051) in multivariate analysis of ENT1 protein expression, analyzed by immunohistochemistry and covariates (Table 5).

Clinical–pathological factors (especially N0/N1, R0/R1) have been widely discussed in terms of patients’ prognosis [49,50,51]. Similarly to Fujita et al. [12], we observed an association of R0 with improved DSS (Figure 1). However, other factors, including primary tumor stage, ASA score, age, gender, and type of resection, did not exhibit an association with DSS (Figure 1).

Like in [12], all patients included in our study received more than three cycles of full-dose chemotherapy. However, the number of patients who refused to continue with or were advised to terminate chemotherapy by an oncologist was 49.3%, which is a higher proportion when compared with data reported from clinical trials [9,10,52]. Clinical–pathologic characteristics of our cohort, patients’ chemotherapy intolerability (24.6%), and/or disease progression (20.2%) might explain the observed overall short DSS median (21 months), whereas overall survival longer than 23 months has been reported for patients on GEM therapy, irrespective of ENT1 expression [9,12,24]. Moreover, in our study, patients with expression of *ENT1* above the median or above 75% demonstrated DSS of 18 and 17 months, respectively, whereas in other studies, 23 [12] and 25.7 [19] months were reported.

Regarding expression of analyzed molecules in PDAC, there is no strictly defined cut-off value to distinguish patients with high and low expression [26]. Only relative values are available, but they differ in the procedure of calculation [12,19]. In Fujita et al. (2010), mRNA levels of a target gene were normalized by expression of the *B2M* housekeeping gene [12], whereas in the study by Giovannetti et al. (2006), values of gene expression were calculated by the *GAPDH* housekeeping/target gene ratio [19]. Importantly, in both studies, information about the stability of these housekeeping genes across the sample cohort was lacking, which complicates interpretation of the data [53]. In our experimental setting, we used the absolute PCR quantification with linear vector with cloned DNA sequence, which is adopted in preferred procedures [54,55,56]. Therefore, it was not possible to compare our values of expression with previously published ones.

Considering our rigorous approach of analysis, the number of observed subjects in the cohort, and the fact that surgery was performed in the high-volume center by specialist surgeons [57,58,59], we hypothesize that the prognostic value of *ENT1*/ENT1 and miR-21 was not confirmed because the cohort contained (i) a relatively high proportion of R1, N1, and T(3,4) patients, (ii) a high number of patients who prematurely terminated GEM therapy, (iii) a high proportion of patients expressing markers or bearing a gene expression signature linked to pancreatic cancer disease progression [60,61,62,63], factors ignored in all previous studies, and/or (iv) a high proportion of patients with low *ENT1* and high miR‑21.

Regarding the last point, we did not demonstrate a significant difference in medians of *ENT1* gene expression among patients with negative, low, and high protein expression, as stratified using the histoscore (Figure 5). A median higher than 100 transcripts of *ENT1*/50 ng RNA in samples collected from negative ENT1 PDAC and approximately 300 transcripts of *ENT1*/50 ng RNA in high ENT1 PDAC indicated that *ENT1* gene expression was not fully proportional to levels of cytoplasmic membrane-embedded ENT1 protein. Posttranscriptional and/or posttranslational regulation [64] and/or altered subcellular localization may play a role in this phenomenon [65,66,67].

Miyamoto et al. (2003) reported upregulated expression of *NOTCH3* in resected PDAC samples [68]. This finding was subsequently confirmed by Vo et al. (2011) and very recently by Song et al. (2018) [35,69]. Several studies have described upregulation of miR-21 in PDAC [25,27,32,36,37]. However, to date, it has not been demonstrated whether *ENT1*/ENT1 is generally downregulated in PDAC. Our results confirmed an overall increase in expression of *NOTCH3* and miR-21 in tumor tissue when compared with normal pancreas (Figure 7B,C). Deregulation of *miRNAs* has been associated with cell growth, promotion of metastatic phenotype, and/or chemoresistance in PDAC. Upregulation of miR‑21 is particularly linked to promotion of cell proliferation, invasion, chemoresistance, and escape from apoptosis [22,70]. Although evidence acquired using hepatocytes and placental cells has suggested constitutive expression of *ENT1* [71,72,73], we found decreased *ENT1* expression in PDAC (Figure 7A). When dividing the cohort based on medians of expression in normal pancreas we observed upregulation of *NOTCH3* only in the subgroup, with low expression of *NOTCH3* below median (<50%) in normal pancreas (Figure 7E), while *ENT1* was downregulated in PDAC of patients with *ENT1* expression above the median (>50%) in normal pancreas (Figure 7D). Upregulation of miR-21 was independent of levels in normal pancreas (Figure 7F). Considering both outcomes of previous reports on *ENT1* expression in PDAC [12,19] and our data (Figure 7D), we hypothesize that length of survival depends on an individual patient’s physiological expression of *ENT1* that seems to determine its expression in tumor. Further investigation is, however, needed, because there is the possibility that the expression of *ENT1* and/or *NOTCH3* in normal pancreas was influenced by factors produced by the tumor.

Considerable expression variability was observed for all the analyzed transcripts in both tumor and normal pancreas. Inter-individual differences in expression of analyzed molecules and, in case of *ENT1*, use of the expression assay recognizing almost all the transcript variants of *ENT1* (Hs01085704_g1) may explain this phenomenon [70,74].

## 4. Materials and Methods

### 4.1. Patients and Pancreatic Cancer Staging

Analysis of *ENT1*, *NOTCH3*, and miR-21 was performed in FFPE samples collected from 69 patients with PDAC who underwent surgical resection between 2006 and 2016 at the Department of Surgery, University Hospital, Hradec Kralove [57,58,59] and showed no substantial postoperative complications. Pancreatic cancer primary tumor/regional lymph nodes/distant metastasis (TNM) staging was performed using the American Joint Committee on Cancer (AJCC) 7th edition system [75]. The resection margins were classified as R0 (tumor-free resection) or R1 (microscopic margin involvement) [76]. Patients were given three or more cycles of adjuvant GEM monotherapy at a dose of 1000 mg/m^2^ on days 1, 8, and 15 in six 28 day cycles and were monitored until 31st December 2018. This research was approved by the Ethics Committee of University Hospital Hradec Kralove (reference number 201607 SO2P).

### 4.2. Preparation of Formalin-Fixed Paraffin-Embedded Samples of Pancreas

All resection specimens were routinely histologically processed, that is, fixed in 10% neutral buffered formalin for 24–72 h, grossly described, cut-up, and sampled in a standardized fashion to evaluate relationship of the tumor to individual resection margins [76,77]. Multiple tissue samples were taken from both PDAC and non-neoplastic surrounding tissue for standard histological examination. The material was first embedded in paraffin, then 3 μm tissue sections were cut and stained with hematoxylin and eosin. For immunohistochemical analysis, in each case, one FFPE block from the tumor periphery containing both normal pancreatic parenchyma and neoplastic tissue was selected. In cases where there was available tissue block with normal pancreatic tissue without any tumor structures (usually tissue samples from resection margin), these blocks were selected and used to validate the method. For qRT-PCR analyses, two individual FFPE blocks (one with tumor tissue and the other containing solely normal pancreatic tissue) were selected for each patient. In the case of tumor tissue blocks, a microdissection method was used to remove parts of the tissue containing only stroma without neoplastic cells. Microdissection was performed by the principal pathologist prior to mRNA extraction, as a low percentage of neoplastic cells in the sample could have affected the outcomes of the analysis [19].

### 4.3. Extraction of mRNA from FFPE Samples and Reverse Transcription

For mRNA isolation from FFPE samples, a well-established method for extraction of total nucleic acids from FFPE tissues was used based on the Recover All^TM^ total nucleic acid isolation kit (ThermoFisher Scientific, Waltman, MA, USA) [78]. The purity of the isolated RNA was checked by the *A_260/280_* ratio. RNA (1 µg) was converted into cDNA in 20 µL reaction using the gb reverse transcription kit from Generi Biotech s.r.o. (Hradec Kralove, Czech Republic) for *ENT1* and *NOTCH3* and a special assay for miR-21 (ThermoFisher Scientific, Waltman, MA, USA) in a Bio-Rad T100^TM^ thermal cycler (Hercules, CA, USA), according to the manufacturer’s protocol.

### 4.4. Quantitative Analysis of ENT1, NOTCH3, and miR-21 Expression

qRT-PCR analysis of *ENT1*, *NOTCH3*, and miR-21 expression in FFPE samples of normal and cancer tissues was performed using QuantStudio^TM^ 6 Flex (Thermo Fisher Scientific, Waltham, MA, USA). cDNA (25 ng) was amplified in 5 µL reaction volumes in a 384-well plate using a TaqMan^®^ Universal Master Mix II, no UNG (Thermo Fisher Scientific, Waltham, MA, USA) and predesigned TaqMan^®^ real-time expression assays for *SLC29A1* (*ENT1*, Hs01085704_g1) [71], *NOTCH3* (Hs01128537_m1), and miR-21 (Hs04231424_s1) [25]. The PCR product sizes of each primer pair were 52, 67, and 64, respectively, enabling accurate and sensitive PCR analysis of gene expression in FFPE [79,80,81]. The amplification of each sample was performed in triplicate, applying the following PCR cycling profile: 95 °C for 3 min, followed by 40 cycles at 95 °C for 15 s and 60 °C for 60 s. We used absolute quantification for genes *NOTCH3* and *ENT1*. To determine the number of *SLC29A1* and *NOTCH3* transcripts, calibration was undertaken with a linear vector containing recombinant cDNA [54,55,56] prepared with primers Hs01085704_g1 and Hs01128537_m1 by company Generi Biotech (Hradec Kralove, Czech Republic). For miR‑21, we used arbitrary units calculated as Δ C_T_, that is, expression of miR‑21 normalized by expression of the reference *RNU43* (assay ID 000397 and 001095, respectively, Thermo Fisher Scientific, Waltham, USA) [25,82].

### 4.5. Immunohistochemical Analysis of ENT1 Expression

Levels of ENT1 protein expression were studied using immunohistochemistry in 63 patients with sufficient remaining material for analysis after microdissection for qRT-PCR quantification. Anti-ENT1 antibody 10D7G2 was obtained from Prof. John Mackey (Cross Cancer Institute, University of Alberta, Canada), and detection was performed as recommended in the original protocol [13,18,20,46]. FFPE sections (2 μm thick) were deparaffinized, followed by pretreatment in DAKO pH 9 buffer (Glostrup, Denmark) for 10 min in a microwave (900 W). The antibody was diluted 1:10 and incubated overnight at 4 °C. A DAKO Envision+ kit (Glostrup, Denmark) was used for detection according to the manufacturer’s instructions. Slides were counterstained with hematoxylin.

Immunohistochemical staining was evaluated semiquantitatively by a histoscore, which included evaluation of the proportion of cells/tissue expressing ENT1 and the intensity of staining, similarly to described previously for endometrial cancer [83]. The percentage of the cancer area stained in high-power fields was examined. The staining intensity was graded as 0 (negative), 1 (weak), 2 (moderate), or 3 (strong), whereas the percentage of positive cells examined was scored as 0 (negative), 1 (<10%), 2 (11–50%), or 3 (>50%). The two values were multiplied, and the histoscore (values from 0 to 9) was determined: 0 (negative), values 1–5 (low), values 6–9 (high) [13]. The person evaluating the immunohistochemical slides was blinded of the results of other tests as well as of patients’ outcomes.

### 4.6. Statistical Analyses

Disease-specific survival (DSS) from the date of surgery was assessed by employing the Kaplan–Meier method, and respective subgroups were compared by the log-rank test [84]. A Cox’s proportional hazards multivariate model was used to corroborate any association of clinical and pathological factors and expression of *ENT1*/ENT1, *NOTCH3*, and/or miR-21 with patients’ DSS [19,25,48,84,85]. Differences in the number of transcripts in subpopulations defined by the calculated histoscore were analyzed by the nonparametric Kruskal–Wallis test. Differences in medians of *ENT1*, *NOTCH3*, and/or miR-21 of expressions in normal pancreas and PDAC were determined by the nonparametric unpaired Mann–Whitney test, and the nonparametric Wilcoxon matched-pair signed-rank test was used to evaluate differences in expression of *ENT1*, *NOTCH3*, and miR-21 among subgroups given by the median of expression in normal pancreas (Figure 7D–F). The data were analyzed using SPSS 18.0 and GraphPad Prism 8.0.2. Statistical significance was set at *p* < 0.05.

## 5. Conclusions

In this retrospective study performed on a well-defined cohort of patients with resected PDAC treated with adjuvant GEM monotherapy, we did not confirm high *ENT1*/ENT1 and low miR‑21 as prognostic biomarkers of improved DSS. Low miR-21 demonstrated prognostic value in N0 and T(1,2) patients only. For the first time, we attempted to assess the prognostic value of *NOTCH3* in such a cohort of patients. However, low *NOTCH3* did not show any association with improved DSS. Additionally, we confirmed that N0 patients had longer DSS. Our data do not preclude the potential application of *ENT1*/ENT1 and miR-21 as prognostic biomarkers for resected patients to improve the cost-effectiveness of chemotherapy in the future. However, they indicate that this procedure is not yet ready to be implemented into clinical decision-making processes. Standard procedures of immunohistochemistry staining scoring and qRT‑PCR analyses must be established, and patients’ characteristics other than *ENT1*/ENT1 and miR-21 expression that likely affect patients’ survival should be considered. Additionally, the data of this study suggested that there is a limited proportional dependence between *ENT1* gene expression evaluated by qRT-PCR in FFPE samples and protein levels assessed by immunohistochemistry and that there is likely an increased risk of *ENT1* downregulation in PDAC of patients with higher *ENT1* expression in normal pancreas, *NOTCH3* tends to be increased in PDAC of patients with low expression in normal pancreas, whereas the increase in miR‑21 is independent of levels in normal pancreas. Even when *ENT1* and *NOTCH3* are deregulated in tumors of patients with high and low expression, respectively, they do not reach levels in the normal pancreas of patients with low expression of *ENT1* and high expression of *NOTCH3*. These findings may serve as a cornerstone of future experimental efforts focusing on inter-individual differences in regulation of *ENT1*, *NOTCH3* and miR-21 in PDAC as well as normal pancreas.

## Figures and Tables

**Figure 1 cancers-11-01621-f001:**
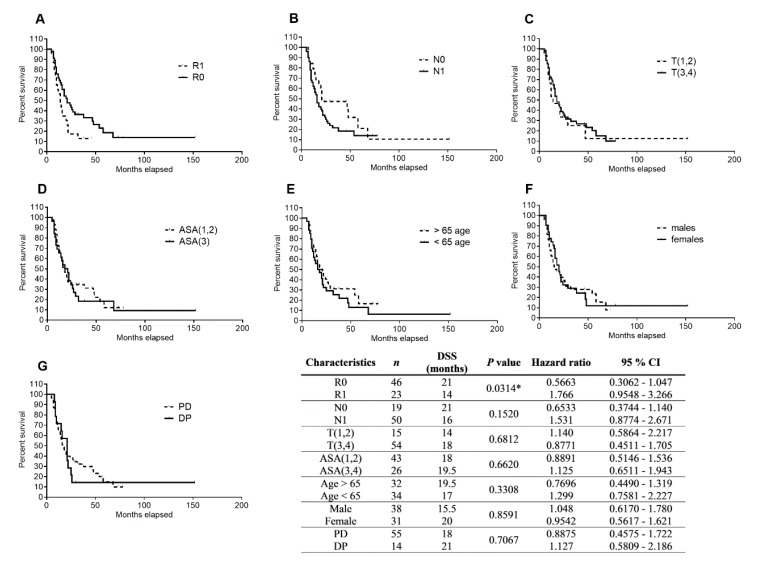
Analysis of patients’ DSS association with resection margin status, regional lymph node involvement, primary tumor stage, ASA score, age, gender, and type of resection using Kaplan–Meier curves. Patients (*n* = 69) were dichotomized based on (**A**) tumor border (R0/R1), (**B**) negative/positive regional lymph nodes (N0/N1), (**C**) primary tumor stage T(1,2)/T(3,4), (**D**) ASA score ASA(1,2)/ASA(3,4), (**E**) patients’ age (> 65 age/< 65 age), (**F**) gender, and (**G**) type of resection (PD/DP). The data were analyzed using the log rank test. Statistical significance is denoted by *, *p* < 0.05.

**Figure 2 cancers-11-01621-f002:**
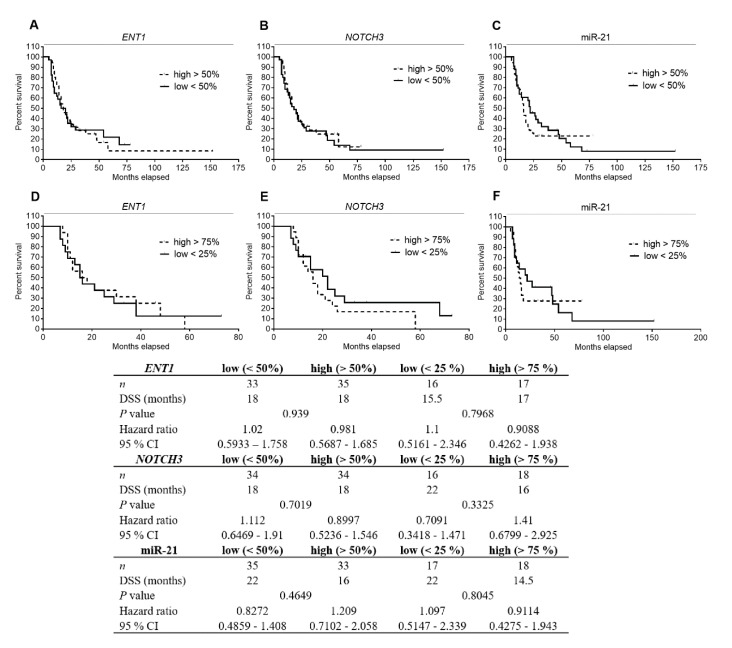
Analysis of patients’ DSS association with *ENT1*, *NOTCH3* mRNA, and miR-21 levels in PDAC tissue using Kaplan–Meier curves. Patients (*n* = 69) were divided into two groups according to the median (low <50%, high >50%) (**A**–**C**) or, alternatively, only subgroups of patients belonging to the first and fourth quartiles (low <25%, high >75%) (**D**–**F**) of expression of *ENT1* (**A**,**D**), *NOTCH3* (**B**,**E**), or miR-21 (**C**,**F**) were selected. No significant differences between the tested subgroups of patients were observed. Statistical significance was evaluated by applying log-rank test analysis.

**Figure 3 cancers-11-01621-f003:**
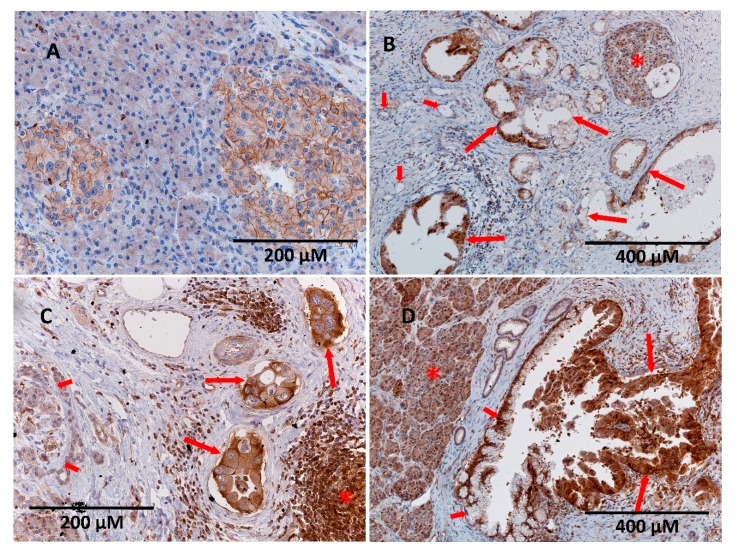
Variability of expression of ENT1 transporter in normal pancreas and pancreatic ductal adenocarcinoma (PDAC). (**A**) Membranous expression is seen in cells of normal Langerhans islets, whereas pancreatic acini are completely negative. (**B**) In some cases, heterogeneous positivity of neoplastic cells of PDAC could be observed (long arrows). Note that the non-neoplastic ducts are negative (short arrows). Membranous positivity in the cells of normal Langerhans islets served as an internal positive control (asterisk). (**C**) PDAC cells with strong membranous and cytoplasmic staining for ENT1 (long arrows). Lymphoid elements show the same level of positivity (asterisk), whereas non-neoplastic exocrine pancreas is virtually negative (short arrows). (**D**) Strong cytoplasmic staining in both the acini of normal pancreas (asterisk) and neoplastic cells lining the dilated duct (long arrows), whereas non-neoplastic ductal cells show only weak membranous, predominantly basal, positivity (short arrows). Original magnification was 200× (**A**,**C**) and 100× (**B**,**D**).

**Figure 4 cancers-11-01621-f004:**
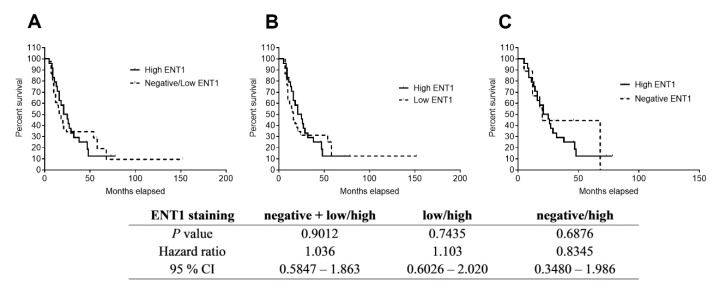
Kaplan–Meier curves showing differences in DSS based on immunohistochemical expression of ENT1. Patients (*n* = 63) were divided according to histoscore (see Methods section) into three groups with negative (*n* = 9), low (*n* = 30), and high (*n* = 24) protein expression of ENT1. Statistical significance was evaluated by applying log-rank test analysis between subgroups with (**A**) high and negative/low ENT1 expression, (**B**) subgroups with high and low ENT1 expression, and (**C**) subgroups with high and negative ENT1 expression. No significant association of ENT1 expression with patients’ DSS was observed.

**Figure 5 cancers-11-01621-f005:**
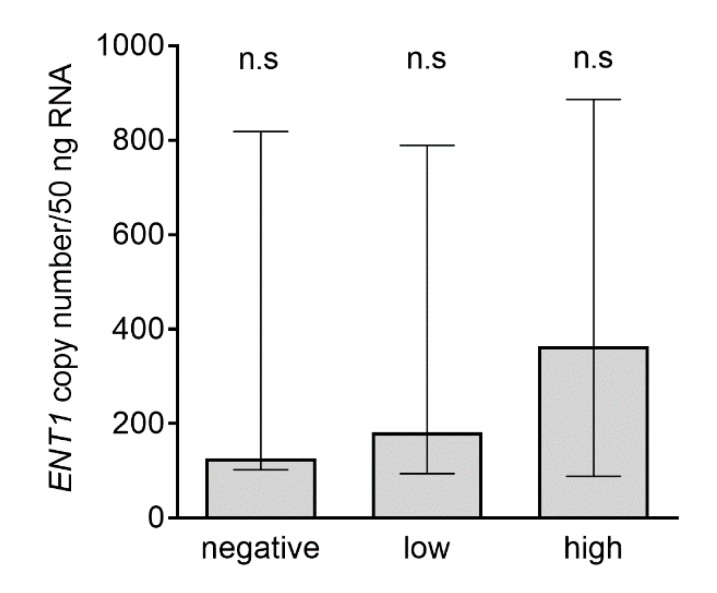
Analysis of *ENT1* transcripts in subgroups with negative, low, and high ENT1 protein expression. Patients were divided based on histoscore of ENT1 expression (and denoted negative (*n* = 9), low (*n* = 30) or high (*n* = 24)); see Methods section. Data are presented as median with interquartile range. Differences in the number of transcripts among subpopulations were analyzed using the nonparametric Kruskal–Wallis test, followed by Dunn’s multiple comparison; n.s., not significant.

**Figure 6 cancers-11-01621-f006:**
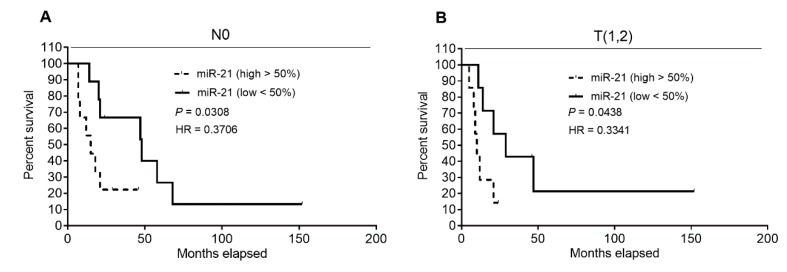
Analysis of patients’ DSS association with miR-21 levels in subgroups with N0 and T(1,2) tumors using Kaplan–Meier curves. Patients in the N0 (**A**) and T(1,2) (**B**) subgroups were divided according to the median (low <50%, high >50%) of expression of miR-21. Low expression of miR-21 was significantly associated with improved DSS in both tested cohorts. Statistical significance was evaluated by applying log-rank test analysis, reaching *p* = 0.0308 for N0 patients and *p* = 0.0438 for patients with T(1,2) tumors.

**Figure 7 cancers-11-01621-f007:**
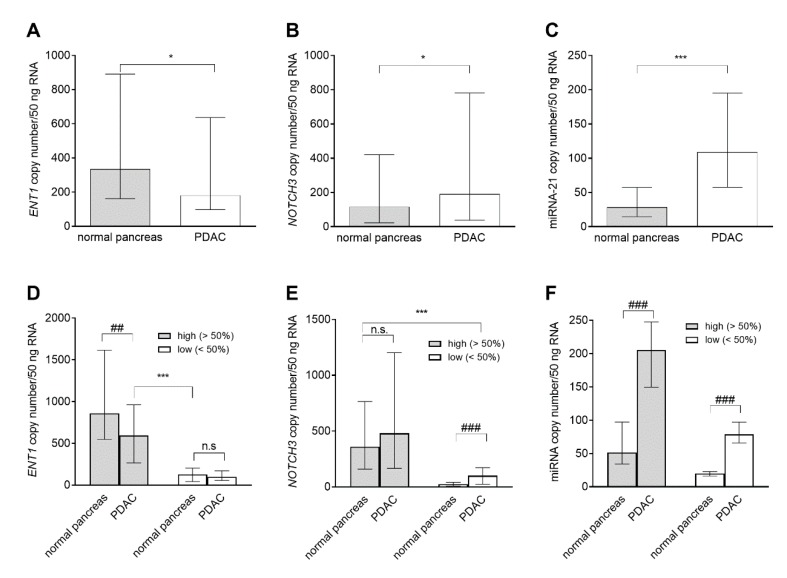
Comparison of mRNA expression of *ENT1, NOTCH3* mRNA, and miR-21 in PDAC and normal pancreas. We compared overall expression between normal pancreas and tumor tissues in the whole sample cohort (**A**–**C**) and subsequently, in subpopulations (high >50% and low <50%) sorted according to the median of the respective molecule expression in healthy tissue (**D**–**F**). The data are presented as median of copy number/50 ng of total RNA with the interquartile range. The nonparametric unpaired Mann–Whitney test was used to evaluate differences in overall expression between the PDAC and normal pancreas (**A**–**C**), *n* = 65, and to compare overall expression in respective subgroups (**D**–**F**); significance was denoted * *p* < 0.05 and *** *p* < 0.001. Statistical significance between expression in PDAC and normal pancreas in subgroups defined by medians (**D**–**F**) was evaluated using the Wilcoxon matched-pair signed-rank test: *n* = 37 (high >50%) and *n* = 27 (low <50%) for *ENT1* (**D**); *n* = 40 (high >50%) and *n* = 34 (low <50%) for *NOTCH3* (**E**), and *n* = 31 (high >50%) and *n* = 33 (low <50%) for miR-21 (**F**); ## *p* < 0.01 and ### *p* < 0.001.

**Table 1 cancers-11-01621-t001:** Clinical–pathological characteristics of patients.

Number of Patients	69
Gender (females/males)	31/38
Age (years)	
Median	65
Range	39–80
Surgery (type of resection)	
PD	55
DP	14
Resection margin status	
R0	46
R1	23
T: stage of primary tumor	
T1	3
T2	12
T3	53
T4	1
N: regional lymph nodes	
N0	19
N1	50
M: distant metastasis	
M0	68
M1	1
DSS from surgery (months)	
Median	21
Range	5–152
ASA (I-III)	
I	1
II	42
III	26

ASA, American Society of Anesthesiologists score; DSS, disease-specific survival; GEM, gemcitabine; DP, distal pancreatectomy; PD, pancreaticoduodenectomy.

**Table 2 cancers-11-01621-t002:** Results of multivariate analysis (Cox’s proportional hazards model) of the effects of clinical and pathological characteristics and *ENT1* mRNA expression on patients’ DSS.

Patient Stratification	*p* Value	Hazard Ratio	95.0% CI	
Gender (male)	0.918	0.970	0.543	1.731
*ENT1* (above median)	0.821	0.934	0.514	1.695
Resection margin status (R1)	0.135	0.624	0.336	1.159
ASA score				
ASA III (reference value)	0.601			
ASA I	0.421	2.395	0.286	20.063
ASA II	0.639	0.869	0.484	1.561
Primary tumor stage				
T4 (reference value)	0.272			
T1	0.434	0.354	0.026	4.786
T2	0.182	0.212	0.022	2.068
T3	0.106	0.169	0.019	1.463
Regional lymph nodes (N1)	0.149	0.591	0.289	1.207
Resection type (DP)	0.675	1.159	0.582	2.309

ASA, American Society of Anesthesiologists score; CI, confidence interval; DP, distal pancreatectomy. Reference levels are displayed between brackets.

**Table 3 cancers-11-01621-t003:** Results of multivariate analysis (Cox’s proportional hazards model) of the effects of clinical and pathological characteristics and *NOTCH3* mRNA expression on patients’ DSS.

Patient Stratification	*p* Value	Hazard Ratio	95.0% CI	
Gender (male)	0.956	0.984	0.553	1.751
*NOTCH3* (above median)	0.383	1.286	0.731	2.263
Resection margin status (R1)	0.126	0.614	0.328	1.147
ASA score				
ASA III (reference value)	0.725			
ASA I	0.484	2.120	0.259	17.379
ASA II	0.773	0.915	0.501	1.671
Primary tumor stage				
T4 (reference value)	0.299			
T1	0.507	0.424	0.034	5.358
T2	0.267	0.275	0.028	2.686
T3	0.144	0.200	0.023	1.728
Regional lymph nodes (N1)	0.152	0.598	0.296	1.209
Resection type (DP)	0.661	1.163	0.593	2.283

ASA, American Society of Anesthesiologists score; CI, confidence interval; DP, distal pancreatectomy. Reference levels are displayed between brackets.

**Table 4 cancers-11-01621-t004:** Results of multivariate analysis (Cox’s proportional hazards model) of the effects of clinical and pathological characteristics and miR-21 expression on patients’ DSS.

Patient Stratification	*p* Value	Hazard Ratio	95.0% CI	
Gender (male)	0.817	0.935	0.530	1.649
miR-21 (above median)	0.089	0.475	0.201	1.120
Resection margin status (R1)	0.113	0.604	0.324	1.126
ASA score				
ASA III (reference value)	0.587			
ASA I	0.477	2.138	0.263	17.375
ASA II	0.530	0.828	0.460	1.491
Primary tumor stage				
T4 (reference value)	0.091			
T1	0.280	0.237	0.017	3.234
T2	0.088	0.131	0.013	1.355
T3	0.036	0.085	0.009	0.846
Regional lymph nodes (N1)	0.175	0.619	0.310	1.238
Resection type (DP)	0.658	1.169	0.586	2.335

ASA, American Society of Anesthesiologists score; CI, confidence interval; DP, distal pancreatectomy. Reference levels are displayed between brackets.

**Table 5 cancers-11-01621-t005:** Results of multivariate analysis (Cox’s proportional hazards model) of the effect of clinical and pathological characteristics and ENT1 protein levels analyzed by immunohistochemistry in FFPE samples on patients’ DSS.

Patient Stratification	*p* Value	Hazard Ratio	95.0% CI	
Gender (male)	0.836	0.936	0.500	1.752
ENT1				
High (reference value)	0.277			
Negative	0.471	1.469	0.517	4.174
Low	0.109	1.800	0.877	3.695
Resection margin status (R1)	0.051	0.505	0.254	1.003
ASA score				
ASA III (reference value)	0.592			
ASA I	0.768	1.385	0.159	12.091
ASA II	0.367	0.747	0.396	1.408
Primary tumor stage				
T4 (reference value)	0.179			
T1	0.637	0.513	0.032	8.162
T2	0.175	0.211	0.022	2.001
T3	0.075	0.138	0.016	1.222
Regional lymph nodes (N1)	0.049	0.424	0.180	0.996
Resection type (DP)	0.865	1.070	0.492	2.325

ASA, American Society of Anesthesiologists score; DP, CI, confidence interval; distal pancreatectomy. Reference levels are displayed between brackets.

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
