# Peer review of "Are ENT1/ENT1, NOTCH3, and miR-21 Reliable Prognostic Biomarkers in Patients with Resected Pancreatic Adenocarcinoma Treated with Adjuvant Gemcitabine Monotherapy?"

_cancers, 2019, doi:10.3390/cancers11111621_

Round 1

Reviewer 1 Report

In the manuscript cancers-605120, the authors studied the prognostic value of ENT1, mir-21 and NOTCH3 in a cohort of patients with resected pancreatic adenocarcinoma. They used qRT-PCR for all the putative biomarkers and IHQ only for ENT1. Results showed that they are not reliable biomarkers of DSS in gemcitabine treated patients, at least under the conditions of this study.

The authors stated several times that is the first time that NOTCH3 is studied as biomarker in resected pancreatic cancer patients for the first time. In fact, this is one of the strong points of the paper due ENT1 and miR-21 had been previously studied. However, I have some concerns regarding this point that must be clarified. They do refer to the first time by means of qRT-PCR? If I’m not wrong, using IHQ, NOTCH3 has been studied as a biomarker in resected pancreatic cancer by Mann CD et al (Ref 32 in the manuscript). Moreover, Miyamoto Y et al (Ref 67 in the manuscript) studied the relation of NOTCH3 in resected pancreatic cancer by qRT-PCR and IHQ, although it’s true that in a small group.

The use of absolute qRT-PCR is a good contribution of the authors to determine levels of ENT1. This could allow the comparison of the results obtained in future cohort analysis. However, unfortunately results showed a high variability. It could be possible to reduce this variability increasing the amount of RNA?

ENT1 have several identified transcript variants according to Genbank. Considering the probe chosen to detect ENT1 which is able to recognize almost all the variants this could be a putative cause for the high variability. Analysis of ENT1 levels with other probe only able to detect longer transcripts could reduce variability.

Considering that the analysis of ENT1, NOTCH3 and miR-21 was performed in paired samples I’m not really sure if the analysis performed in figure 7 D, E and F is contributing to the study and the description in the written manuscript is hard to follow.

Minor:

Results section:

Page 5, line 120: Figure 5A should be Figure 1A.

In general, is difficult to identify to what group correspond the line in Kaplan-Meier curves. Please, show it clearer in figures 2, 4 and 6.

Page 9, line 179: ‘Figure 4’ shouldn’t appear.

Page 11, line 203: the legend used for regional lymph node should include N1/N0 together with positivity/negativity.

Materials and Methods section:

Section 4.2: Page 18, line 406: What do the authors mean when they say ‘Tissue samples of normal pancreas were used to validated the method’?

What was the criteria used to select the section of non-tumoural pancreas to perform qRT-PCR?

Author Response

In the manuscript cancers-605120, the authors studied the prognostic value of ENT1, mir-21 and NOTCH3 in a cohort of patients with resected pancreatic adenocarcinoma. They used qRT-PCR for all the putative biomarkers and IHQ only for ENT1. Results showed that they are not reliable biomarkers of DSS in gemcitabine treated patients, at least under the conditions of this study.

The authors stated several times that is the first time that NOTCH3 is studied as biomarker in resected pancreatic cancer patients for the first time. In fact, this is one of the strong points of the paper due ENT1 and miR-21 had been previously studied. However, I have some concerns regarding this point that must be clarified. They do refer to the first time by means of qRT-PCR? If I’m not wrong, using IHQ, NOTCH3 has been studied as a biomarker in resected pancreatic cancer by Mann CD et al (Ref 32 in the manuscript). Moreover, Miyamoto Y et al (Ref 67 in the manuscript) studied the relation of NOTCH3 in resected pancreatic cancer by qRT-PCR and IHQ, although it’s true that in a small group.

Mann C.D. et al. and Miyamoto Y. et al. analyzed NOTCH3 protein and gene expression, respectively, in PDAC resected patients. However, the tested cohorts were not defined in terms of pharmacotherapy and/or radiotherapy. Type of pharmacotherapy (no pharmacotherapy, gemcitabine, 5-fluorouracil, folfirinox, or another therapy was used to treat the resected patients?) and/or radiotherapy application may have crucial impact on association between expression of selected molecules and patients’ overall survival.

In our study we performed the analysis in the well-defined cohort of patients. Therefore, we declared in the manuscript, that “we, for the first time, analyzed NOTCH3 expression in FFPE samples collected from a homogeneous group of patients with resected PDAC treated with adjuvant GEM therapy (n = 69) to evaluate the prognostic value of the associated transcripts for estimation of disease specific survival (DSS)”. Nevertheless, to avoid misinterpretation, we omitted the phrase “for the first time” from Abstract and Introduction of the revised manuscript.

The use of absolute qRT-PCR is a good contribution of the authors to determine levels of ENT1. This could allow the comparison of the results obtained in future cohort analysis. However, unfortunately results showed a high variability. It could be possible to reduce this variability increasing the amount of RNA?

We used 1 µg RNA per 20 µl reaction for reverse transcription, i.e., the highest amount of RNA recommended by the manufacturer (Generi-Biotech, s.r.o.). This information was added into the revised manuscript (line 425). It is also important to note that we are not the first reporting on high gene expression variability of ENT1 in tumor cells and normal tissue [1].

ENT1 have several identified transcript variants according to Genbank. Considering the probe chosen to detect ENT1 which is able to recognize almost all the variants this could be a putative cause for the high variability. Analysis of ENT1 levels with other probe only able to detect longer transcripts could reduce variability:

We agree with the reviewer that expression variability can be potentially reduced by use of assay detecting only longer transcripts [2,3]. However, specific transcripts of ENT1 (most of them encodes the same protein) associated with improved survival in patients with adjuvant GEM therapy have not been described, therefore, we used expression assay recognizing almost all ENT1 transcript variants. Moreover, we had to select expression assay producing short amplicons, which increases reliability of PCR analysis of gene expression performed in FFPE samples (expression assay Hs01085703_g1 produced by Thermo Fisher Scientific provides amplicons of 52 bp length).

To address the reviewer’s comment, we added information into the Discussion section describing potential contribution of the used expression assay to ENT1 expression variability “Considerable expression variability was observed for all the analyzed transcripts in both tumor and normal pancreas. Inter-individual differences in expression of analyzed molecules and, in case of ENT1, use of the expression assay recognizing almost all the transcript variants of ENT1 (Hs01085704_g1) may explain this this phenomenon [3,4].“ Please, see lines 390-393.

Considering that the analysis of ENT1, NOTCH3 and miR-21 was performed in paired samples I’m not really sure if the analysis performed in figure 7 D, E and F is contributing to the study and the description in the written manuscript is hard to follow.

Based on the reviewer’s comment we reconsidered the statistical approach used to evaluate the difference between tested groups. In the revised manuscript we are using the non-parametric and unpaired Mann-Whitney test to compare the overall expression of ENT1, NOTCH3 and miR-21 in PDAC and normal tissue. Because we hypothesized that extent of tumor deregulation of ENT1, NOTCH3 and miR-21 might be dependent on their levels in corresponding normal pancreas, we divided patients based on the median of expression of each marker in healthy pancreas (high, > 50%; low, < 50 %) and using Wilcoxon matched-pair signed rank test we compared the expression of each marker in both types of tissue in these subgroups.

Our findings might be of interest for scientific community as they indicate that there is an increased risk of ENT1 downregulation in PDAC of patients with higher ENT1 expression in normal pancreas and NOTCH3 tends to be increased in PDAC of patients with low expression in normal pancreas. Upregulation of miR-21 seems to be independent of levels in normal pancreas. In conclusion extent of tumor deregulation of ENT1 and NOTCH3 is dependent on  expression in normal pancreas. Therefore, it can be speculated that the potential resistance to gemcitabine is given by physiological expression rather than gene deregulation in tumor.

Moreover, we did our best to improve legibility of respective parts in the revised manuscript. Please, see changes in Results section (lines 262 – 273), Figure 7 and Figure 7 legend, and corresponding part in Discussion (lines 377-389).

Minor:

Results section:

Page 5, line 120: Figure 5A should be Figure 1A.

It was corrected.

In general, it is difficult to identify to what group correspond the line in Kaplan-Meier curves. Please, show it clearer in figures 2, 4 and 6:

All mentioned figures were remade to improve identification of tested groups and their corresponding lines.

Page 9, line 179: ‘Figure 4’ shouldn’t appear.

This was omitted from the text of the revised manuscript.

Page 11, line 203: the legend used for regional lymph node should include N1/N0 together with positivity/negativity.

N1/N0 was added.

Materials and Methods section:

Section 4.2: Page 18, line 406: What do the authors mean when they say ‘Tissue samples of normal pancreas were used to validated the method’? What was the criteria used to select the section of non-tumoural pancreas to perform qRT-PCR?:

To improve clarity of this section we reworded it as follows: “For immunohistochemical analysis, in each case, one FFPE block from the tumor periphery containing both normal pancreatic parenchyma and neoplastic tissue was selected. In cases, where there was available tissue block with normal pancreatic tissue without any tumor structures (usually tissue samples from resection margin), these blocks were selected and used to validate the method”. See lines 410 -414.

References:

Farrell, J.J.; Elsaleh, H.; Garcia, M.; Lai, R.; Ammar, A.; Regine, W.F.; Abrams, R.; Benson, A.B.; Macdonald, J.; Cass, C.E., et al. Human equilibrative nucleoside transporter 1 levels predict response to gemcitabine in patients with pancreatic cancer. Gastroenterology 2009, 136, 187-195, doi:10.1053/j.gastro.2008.09.067. Alemu, E.Y.; Carl, J.W., Jr.; Corrada Bravo, H.; Hannenhalli, S. Determinants of expression variability. Nucleic Acids Res 2014, 42, 3503-3514, doi:10.1093/nar/gkt1364. Hulse, A.M.; Cai, J.J. Genetic variants contribute to gene expression variability in humans. Genetics 2013, 193, 95-108, doi:10.1534/genetics.112.146779. Gnoni, A.; Licchetta, A.; Scarpa, A.; Azzariti, A.; Brunetti, A.E.; Simone, G.; Nardulli, P.; Santini, D.; Aieta, M.; Delcuratolo, S., et al. Carcinogenesis of pancreatic adenocarcinoma: precursor lesions. Int J Mol Sci 2013, 14, 19731-19762, doi:10.3390/ijms141019731.

Reviewer 2 Report

The manuscript is an original study on the role of prognostic biomarkers in resected PDAC patients treated with adjuvant Gemcitabine. In particular authors investigated the intrigue role of ENT1/ENT1 and miR-21 in this setting, even if their evaluation is not ready to be implemented into clinical decision-making processes. Moreover, PDAC sensitivity to gemcitabine is potentially mediated by ENT1 or NOTCH3 related to individual physiological patient expression.

To improve the quality of the manuscript, only minor revisions are required:

English language should be revised; Abbreviations should be checked (for example in line 51 mFOLFIRINOX); Authors should revised some misspelling in the manuscript; In figure 3 legend, Authors should specify the magnification of captured pictures; In the Discussion, Authors should discuss of the crucial role of miRNA in pancreatic cancer (please refer to follow manuscripts: a) MicroRNA in pancreatic adenocarcinoma: predictive/prognostic biomarkers or therapeutic targets? Brunetti et al. Oncotarget. 2015 Sep 15;6(27):23323-41. Review; b) Carcinogenesis of pancreatic adenocarcinoma: precursor lesions. Gnoni et al. Int J Mol Sci. 2013 Sep 30;14(10):19731-62. doi: 10.3390/ijms141019731. Review).

Author Response

The manuscript is an original study on the role of prognostic biomarkers in resected PDAC patients treated with adjuvant Gemcitabine. In particular authors investigated the intrigue role of ENT1/ENT1 and miR-21 in this setting, even if their evaluation is not ready to be implemented into clinical decision-making processes. Moreover, PDAC sensitivity to gemcitabine is potentially mediated by ENT1 or NOTCH3 related to individual physiological patient expression.

To improve the quality of the manuscript, only minor revisions are required:

English language should be revised.

The English language was edited by the professional service “Sees-editing” specialized on the editing of scientific manuscripts prior submitting to Cancers and we did our best to additionally improve the spelling and wording of the revised manuscript.

Abbreviations should be checked (for example in line 51 mFOLFIRINOX).

We have double checked the abbreviations in the revised manuscript. Explanation of the abbreviation “mFOLFIRINOX” is provided in the revised manuscript as follows “modified-dose FOLFIRINOX (mFOLFIRINOX) consisting of oxaliplatin at 85 mg/m², leucovorin at 400 mg/m², irinotecan at 150 mg/m², and 5-fluoroucil at 2.4 g/m² increases…” (lines 50-52).

Authors should revise some misspelling in the manuscript.

We double checked the spelling to improve quality of the revised manuscript.

In figure 3 legend, Authors should specify the magnification of captured pictures.

We added to the figure 3 legend information about original magnification as follows: Original magnification was 200× (A, C) and 100× (B, D). See lines 194-195.

In the Discussion, Authors should discuss of the crucial role of miRNA in pancreatic cancer (please refer to follow manuscripts: a) MicroRNA in pancreatic adenocarcinoma: predictive/prognostic biomarkers or therapeutic targets? Brunetti et al. Oncotarget. 2015 Sep 15;6(27):23323-41. Review; b) Carcinogenesis of pancreatic adenocarcinoma: precursor lesions. Gnoni et al. Int J Mol Sci. 2013 Sep 30;14(10):19731-62. doi: 10.3390/ijms141019731. Review):

We added the required information and references into the revised manuscript: „Deregulation of miRNAs have been associated with cell growth, promotion of metastatic phenotype and/or chemoresistance in PDAC. Upregulation of miR-21 is particularly linked to promotion of cell proliferation, invasion, chemoresistance and escape from apoptosis [1,2]“. See lines 378-380.

 References:

Brunetti, O.; Russo, A.; Scarpa, A.; Santini, D.; Reni, M.; Bittoni, A.; Azzariti, A.; Aprile, G.; Delcuratolo, S.; Signorile, M., et al. MicroRNA in pancreatic adenocarcinoma: predictive/prognostic biomarkers or therapeutic targets? Oncotarget 2015, 6, 23323-23341, doi:10.18632/oncotarget.4492. Gnoni, A.; Licchetta, A.; Scarpa, A.; Azzariti, A.; Brunetti, A.E.; Simone, G.; Nardulli, P.; Santini, D.; Aieta, M.; Delcuratolo, S., et al. Carcinogenesis of pancreatic adenocarcinoma: precursor lesions. Int J Mol Sci 2013, 14, 19731-19762, doi:10.3390/ijms141019731.

Reviewer 3 Report

The authors stated that ENT1 and NOTCH3 represent biomarkers for gemcitabine sensitivity

The reviewer had some questions to their manuscript. 

1.     At first, their title is appropriate to their conclusion ?

2.     Reviewer did not understand why they wrote gemcitabine sensitivity mediated by ENT1 or NOTH3. They did not prove this in their manuscript at all.

3.     In Fig.1, authors should explain why T3, 4 group survival were longer than T1, 2 group.

4.     In Fig. 6, authors showed miR-21 high/low is associated with prognosis of patients with N0 (N=19)or T(1,2)(N=15). Did they think these sample number are statistically enough to prove statistic significant ?

5.     In Fig. 7, Please explain why they divided normal pancreas samples in two group (high/low) ? Is this meaningful in spite of normal samples ? Moreover, mRNA expressions of normal sample and PDAC are negatively correlated ?

6.      Finally, please add ethics committee approval number in material and methods.    

Author Response

The reviewer had some questions to their manuscript.

At first, their title is appropriate to their conclusion?

Before submission of the manuscript to Cancers we tried to find a title that would reflect explicitly findings of our study. Even though titles were long and complicated, they did not cover the broadness of our results. In conclusion, we consider the title “Are ENT1/ENT1, NOTCH3 and miR‑21 reliable Prognostic Biomarkers in Patients with Resected Pancreatic Adenocarcinoma Treated with Adjuvant Gemcitabine Monotherapy? “ as the most appropriate one, because it reflects our findings in the context of published results in the most straightforward way.

Reviewer did not understand why they wrote gemcitabine sensitivity mediated by ENT1 or NOTCH3. They did not prove this in their manuscript at all.

Thank you for this comment. Words “gemcitabine sensitivity mediated by ENT1 or NOTCH3” are not used in the revised manuscript. In abstract, “sensitivity mediated by ENT1” was removed. Instead we extended the conclusions (in abstract) of our study as follows “We also conclude that occurrence of ENT1 and NOTCH3 deregulation in PDAC is dependent on their expression in normal pancreas” (lines 30-31). We also attempted to improve clarity of results description in abstract (see lines 25-28).

The information „Importantly, based on our data (Figure 7), we suggest that level of GEM sensitivity potentially mediated by ENT1 or NOTCH3 depends on an individual patient’s physiological expression in the pancreas rather than on tumor deregulation” in Discussion section was rephrased in the revised manuscript and now reads “Considering both outcomes of previous reports on ENT1 expression in PDAC [1,2] and our data (Figure 7D), we hypothesize that length of survival depends on an individual patient’s physiological expression of ENT1 that seems to determine its expression in tumor.” (lines 390-393 in the revised manuscript).

In Fig.1, authors should explain why T3, 4 group survival were longer than T1, 2 group.

In Fig. 1 of the submitted manuscript we showed that there was no statistically significant difference between T1,2 and T3,4 groups. Therefore, it was not possible to conclude that T3,4 group survival was longer than in T1,2 group. This conclusion did not appear in both the original submission and the revised manuscript. Lack of significance (or why T3,4 group tended to longer survival) can be explained by the fact that T3,4 group (n = 54) involved high proportion of patients with clinical-pathological characteristics associated with improved survival: R0 (35/54), N0 (11/54), and R0/N0 (9/54), while T1/2 group (n = 15) included relatively high number of patients with unfavorable clinical-pathological factors R1 (4/15), N1 (6/15), R1/N1 (2/15).

In Fig. 6, authors showed miR-21 high/low is associated with prognosis of patients with N0 (N=19) or T(1,2)(N=15). Did they think these sample number are statistically enough to prove statistic significant?

The sample sizes of these subgroups were small but comparable to that used by e.g. Spratlin et al. (n = 21; palliative GEM-based regimen) [3]. Even when the sample sizes were small, the performed log-rank tests clearly demonstrated statistical significance. Moreover, our data from N0 subgroup correlates with the data published by Dillhoff et al. [4].

In Fig. 7, Please explain why they divided normal pancreas samples in two group (high/low)? Is this meaningful in spite of normal samples? Moreover, mRNA expressions of normal sample and PDAC are negatively correlated?

Based on the reviewer’s comment we reconsidered the statistical analysis used to evaluate the difference between tested groups. In the revised manuscript we are using non-parametric Mann-Whitney test to compare the overall expression of ENT1, NOTCH3 and miR-21 in PDAC and normal tissue. Our data on ENT1 and miR-21 correlated with the reported ones [4-7] and we, for the first time, showed that overall expression of ENT1 is decreased in PDAC. Because we hypothesized that extent of ENT1, NOTCH3 and miR-21 tumor deregulation might be dependent on their levels in corresponding normal pancreas, we divided patients based on the median of expression of each marker in normal pancreas (high, > 50%; low, < 50 %) and using Wilcoxon matched-pair signed rank test we compared the expression of each marker in both types of tissue in these subgroups (high, > 50%; low, < 50 %). We showed that tumor ENT1 was downregulated when compared with normal pancreas of high (> 50%) subgroup, while there was comparable expression of ENT1 in tumor and normal pancreas of low (< 50 %) subgroup. Overall expression of ENT1 in PDAC of high (> 50%) subgroup was higher than that found in normal tissue of low (< 50 %) subgroup (analyzed by Mann-Whitney test). In addition, tumor NOTCH3 was upregulated when compared with its levels in normal pancreas of low (< 50%) subgroup, while there was comparable expression of NOTCH3 in tumor and normal pancreas of high (> 50%) subgroup. Mann-Whitney test showed that levels of NOTCH3 in PDAC of low (< 50%) subgroup was lower than those found in normal tissue of high (> 50%) subgroup. It indicates that there is an increased risk of ENT1 downregulation in PDAC of patients with higher ENT1 expression in normal pancreas and NOTCH3 tends to be increased in PDAC of patients with low expression in normal pancreas. Upregulation of miR-21 seems to be independent of levels in normal pancreas.

To address the second part of the question, we observed negative correlation of i) ENT1 mRNA expression in PDAC and normal pancreas in patients belonging to high (> 50%) subgroup and ii) NOTCH3 in low (< 50%) subgroup.

We consider these findings very important as they indicate that extent of deregulation of tumor ENT1 and NOTCH3 is dependent on expression in normal pancreas. Therefore, based on both reported data [1,2] and our results it can be speculated that length of survival depends on an individual patient’s physiological expression of ENT1 that seems to determine its expression in tumor.

Please, see changes in Results section (lines 262 – 273), Figure 7 and Figure 7 legend, and corresponding part of Discussion (lines 377-389).

Finally, please add ethics committee approval number in material and methods.

Reference number of Ethics committee approval is 201607 SO2P and it was provided on page 18, line 404 of the original manuscript (paragraph 4.1). For the purpose of the reviewing process, we provide the reviewer with the copy of the original approval and blank copy of informed consent.

References:

Fujita, H.; Ohuchida, K.; Mizumoto, K.; Itaba, S.; Ito, T.; Nakata, K.; Yu, J.; Kayashima, T.; Souzaki, R.; Tajiri, T., et al. Gene expression levels as predictive markers of outcome in pancreatic cancer after gemcitabine-based adjuvant chemotherapy. Neoplasia 2010, 12, 807-817. Giovannetti, E.; Del Tacca, M.; Mey, V.; Funel, N.; Nannizzi, S.; Ricci, S.; Orlandini, C.; Boggi, U.; Campani, D.; Del Chiaro, M., et al. Transcription analysis of human equilibrative nucleoside transporter-1 predicts survival in pancreas cancer patients treated with gemcitabine. Cancer Res 2006, 66, 3928-3935, doi:10.1158/0008-5472.CAN-05-4203. Spratlin, J.; Sangha, R.; Glubrecht, D.; Dabbagh, L.; Young, J.D.; Dumontet, C.; Cass, C.; Lai, R.; Mackey, J.R. The absence of human equilibrative nucleoside transporter 1 is associated with reduced survival in patients with gemcitabine-treated pancreas adenocarcinoma. Clin Cancer Res 2004, 10, 6956-6961, doi:10.1158/1078-0432.CCR-04-0224. Dillhoff, M.; Liu, J.; Frankel, W.; Croce, C.; Bloomston, M. MicroRNA-21 is overexpressed in pancreatic cancer and a potential predictor of survival. J Gastrointest Surg 2008, 12, 2171-2176, doi:10.1007/s11605-008-0584-x. Mann, C.D.; Bastianpillai, C.; Neal, C.P.; Masood, M.M.; Jones, D.J.; Teichert, F.; Singh, R.; Karpova, E.; Berry, D.P.; Manson, M.M. Notch3 and HEY-1 as prognostic biomarkers in pancreatic adenocarcinoma. PLoS One 2012, 7, e51119, doi:10.1371/journal.pone.0051119. Miyamoto, Y.; Maitra, A.; Ghosh, B.; Zechner, U.; Argani, P.; Iacobuzio-Donahue, C.A.; Sriuranpong, V.; Iso, T.; Meszoely, I.M.; Wolfe, M.S., et al. Notch mediates TGF alpha-induced changes in epithelial differentiation during pancreatic tumorigenesis. Cancer Cell 2003, 3, 565-576. Vychytilova-Faltejskova, P.; Kiss, I.; Klusova, S.; Hlavsa, J.; Prochazka, V.; Kala, Z.; Mazanec, J.; Hausnerova, J.; Kren, L.; Hermanova, M., et al. MiR-21, miR-34a, miR-198 and miR-217 as diagnostic and prognostic biomarkers for chronic pancreatitis and pancreatic ductal adenocarcinoma. Diagn Pathol 2015, 10, 38, doi:10.1186/s13000-015-0272-6.

Round 2

Reviewer 1 Report

In the revised manuscript cancers-605120, the authors have satisfactory addressed the points raised in the previous revision.

Nevertheless, I have one more concern. It’s true that their results showed that the expression of ENT1 and NOTCH3 in the paired non-cancerous sample of the pancreas could define the risk of the patients. However, authors don’t compare the levels of the surrounding tissue with the levels of normal pancreas without any sign of cancer and the levels detected on the surrounding tissues do not have to be normal since they can be influenced by several factors related to the tumor even when the cells appear healthy. This should be included at least in the discussion to avoid misunderstanding.

Author Response

In the revised manuscript cancers-605120, the authors have satisfactory addressed the points raised in the previous revision. Nevertheless, I have one more concern. It’s true that their results showed that the expression of ENT1 and NOTCH3 in the paired non-cancerous sample of the pancreas could define the risk of the patients. However, authors don’t compare the levels of the surrounding tissue with the levels of normal pancreas without any sign of cancer and the levels detected on the surrounding tissues do not have to be normal since they can be influenced by several factors related to the tumor even when the cells appear healthy. This should be included at least in the discussion to avoid misunderstanding.

We agree with the reviewer. Therefore, we included following information into the Discussion of the revised manuscript: “Further investigation is, however, needed, because there is possibility that the expression of ENT1 and/or NOTCH3 in normal pancreas was influenced by factors produced by the tumor.” See lines 389 – 391. Moreover, we added a word “likely” into the Conclusions (line 493; “…that there is likely an increased risk of ENT1 downregulation in PDAC of patients with higher ENT1 expression in normal pancreas”) of the revised manuscript.

Reviewer 3 Report

Thank you for their effort.

This content becomes better than previous one.

But reviewer still think that this negative data is not suitable for cancers, because of no novel message for cancers readers.

Author Response

Thank you for their effort. This content becomes better than previous one. But reviewer still think that this negative data is not suitable for cancers, because of no novel message for cancers readers.

We thank the reviewer for her/his opinion.